# The Investigative Role of Statins in Ameliorating Lower Urinary Tract Symptoms (LUTS): A Systematic Review

**DOI:** 10.3390/jcm10030416

**Published:** 2021-01-22

**Authors:** Giorgio Ivan Russo, Gaetano Larganà, Arcangelo Sebastianelli, Andrea Cocci, Marina Di Mauro, Ilenia Rapallo, Giuseppe Morgia, Matteo Mario Morgia, Sandro La Vignera, Rosita Condorelli, Aldo E. Calogero, Iacopo Olivotto, Simone Morselli, Sergio Serni, Mauro Gacci

**Affiliations:** 1Urology Section, Department of Surgery, University of Catania, 95100 Catania, Italy; g.a.largana@gmail.com (G.L.); marinadimauro@live.it (M.D.M.); ilenia.rapallo@gmail.com (I.R.); Giuseppe.morgia@unict.it (G.M.); matteomorgia@gmail.com (M.M.M.); 2Department of Minimally Invasive and Robotic Urologic Surgery and Kidney Transplantation, University of Florence, 50100 Florence, Italy; arcangelo.sebastianelli@gmail.com (A.S.); cocci.andrea@gmail.com (A.C.); simone.morselli.89@gmail.com (S.M.); Sergio.serni@unifi.it (S.S.); maurogacci@yahoo.it (M.G.); 3Department of Clinical and Experimental Medicine, University of Catania, 95100 Catania, Italy; sandrolavignera@unict.it (S.L.V.); rosita.condorelli@unict.it (R.C.); acaloger@unict.it (A.E.C.); 4Cardiomyopathy Unit, Careggi University Hospital, 50100 Florence, Italy; iacopo.olivotto@unifi.it

**Keywords:** prostate, LUTS, hypercholesterolemia, tryglycerides, metabolic syndrome, prostate enlargement

## Abstract

Previous data have shown that patients with metabolic syndrome (MetS) and lower urinary tract symptoms (LUTS) secondary to benign prostatic enlargement (BPE) could be refractory to the medical treatment. In this context, the evidence suggests a role for statin use in LUTS/BPE patients. The present systematic review aimed to evaluate the impact of statins on the treatment of men with LUTS/BPE. This review has been registered on PROSPERO (CRD42019120729). A systematic review of English-language literature was performed up to January 2020 in accordance with the preferred reporting items for systematic reviews and meta-analyses (PRISMA statement) criteria. Retrieved studies had to include adults with LUTS connected to BPE treated with statins drugs for metabolic syndrome. After removing duplicates, a total of 381 studies were identified by the literature search and independently screened. Of these articles, 10 fit the inclusion criteria and were further assessed for eligibility. Data from our systematic review suggest that a long-term therapy with statins, at least 6 months, is required to achieve significant impacts on prostate tissue and LUTS. Moreover, besides statins’ direct activity, the risk reduction of LUTS might be connected to the improvement of hypercholesterolemia and MetS. The role of statins for the treatment of LUTS/BPE may be beneficial; however, evidence from robust studies is not enough, and more clinical trial are required.

## 1. Introduction

Lower urinary tract symptoms (LUTS) in men are frequently associated with bladder outlet obstruction (BOO) and benign prostatic enlargement (BPE), often due to benign prostatic hyperplasia (BPH) [1]. LUTS are a common complaint in adult men, causing quality of life (QoL) limitations [2] and impairments in social and personal activities [3]. The prevalence of moderate to severe LUTS ranges from 18% to nearly 40% [4].

Different classes of drugs are currently available for the treatment of LUTS/BPH: alpha-blockers, 5-alpha-reductase inhibitors (5-ARI), inhibitors of phosphodiesterase-5 (PDE5-i), anti-cholinergic drugs, and beta-3 adrenergic receptor agonists [1].

Recent papers have shown a significant correlation between metabolic syndrome (MetS) and LUTS/BPE [5,6]. In a cross-sectional study by Pashootan et al., the authors studied the frequency and severity of LUTS in 4666 men, of which 2273 were with MetS. The presence of MetS was associated with higher International Prostate Symptom Score (IPSS) total scores, both voiding and storage scores, and even for each individual question of the IPSS. The percentage of patients with MetS significantly increased with IPSS severity [5].

Moreover, De Nunzio et al., in a 2012 systematic review, evaluated 11 papers about the relationship between BPH and LUTS. The cumulative evidence summarized in this review suggests an association between MetS and its mediators, and the development of BPH [6]. 

MetS is a cluster of metabolic components, including obesity, reduced HDL cholesterol, high blood pressure, high triglyceride and insulin resistance [7]. It is associated with an increased risk of cardiovascular disease, type 2 diabetes and mortality of all causes [8,9]. 

Interestingly, previous data have shown that patients with MetS could be refractory to medical treatments and may exhibit symptoms persistence after surgery [10,11]. In fact, MetS is associated with an increase in intravesical prostatic protrusion (IPP), together with an increase in prostate volume (PV), explaining the lack of satisfaction with medical therapy in those patients with metabolic alterations [10]. Moreover, De Nunzio et al. have shown that MetS and smoking increased the risk of moderate/severe persistent nocturia after the transurethral resection of the prostate (TURP) [11]. 

In this context, the evidence suggests a role for statin use in LUTS/BPE patients [12,13]. Statins are 3-hydroxy-3-methylglutaryl-coenzyme A (HMG-CoA) reductase inhibitors used to improve serum lipid parameters, including reductions in total cholesterol, low-density lipoprotein, apoprotein B and triglycerides [14]. These drugs are prescribed for both the primary and secondary prevention of coronary heart disease, stroke and peripheral artery disease [15]. Statins have apoptotic and anti-inflammatory properties [16]. They are able to reduce the isoprenylation of the G-protein Rho and Ras, which could lead to prostatic smooth muscle relaxation [17], increased apoptosis and the reduced proliferation of prostate stroma cell in vitro [18], with a modulation of connective growth factor expression allowing a reduction in bladder and prostate fibrosis [18]. Figure 1 shows Ras pathways. Moreover, a systematic and meta-analysis review by Schooling et al. [19] showed that statins reduce testosterone levels, and this effect could lead to their protective effects on prostate growth and on the severity of the LUTS. All these factors could produce beneficial effects for LUTS/BPE. Based on all these premises, the following systematic review aimed to evaluate the impact of statins on the treatment of men with LUTS/BPE.

## 2. Experimental Section

A systematic review of English-language literature was performed up to January 2020 in accordance with the preferred reporting items for systematic reviews and meta-analyses (PRISMA statement) criteria [20]. The Medline, Scopus (Elsevier, Amsterdam, The Netherlands), Web of Science (Clarivariate, Philadelphia, PA, USA) and PubMed databases (National Institutes of Health, Bethesda, MD, USA) were screened separately by two different authors using a single query in order to identify the articles describing the use of statins, and their effects on LUTS and BPE. Conflicts were resolved by discussion or with an independent arbiter. The authors screened all the articles indexed in the aforementioned databases using the following query: “Statins” AND (“LUTS” OR “prostate” OR “benign prostatic hyperplasia” OR “lower urinary tract symptoms” OR “benign prostatic enlargement” OR “benign prostatic hyperplasia”) for each database. 

The retrieved studies had to include adults with LUTS connected to BPE treated with statins drugs for metabolic syndrome. 

Randomized clinical trials, retrospective, prospective, observational, single-arm studies, and comparative studies on humans, were included, while case reports and reviews, letters to editors, papers on animals and “full text not available in English” were excluded. According to the predefined inclusion and exclusion criteria, titles and abstracts were screened, and articles categorized. After reading the abstract, a more thorough assessment was performed by looking at the full texts of the paper. References from the included studies were manually retrieved to identify additional studies of interest. A new excel table was built including data from the selected articles and including the numbers of participants, interventions, comparators, outcomes, and study design (PICOS), as indicated by the Systematic Review Guidance of the Centre for Reviews and Dissemination of the University of York (UK) (Centre for Reviews and Dissemination. Guidance for undertaking reviews in health care. www.york.ac.uk/crd/guidance). 

In view of the presence of multiple methods for evaluating and studying BPE and LUTS, the results of each study may differ from each other. However, the authors only included articles with scientifically approved final BPH and LUTS assessment results.

### Risk of Bias Assessment

The risk of bias in all included randomized clinical trials (RCTs) was evaluated according to the Cochrane collaboration risk of bias [21]. For nonrandomized studies, we used the ROBINS-I assessement to address the methodological quality [22]. 

## 3. Results

After removing duplicates, a total of 381 studies were identified by the literature search and independently screened. Of these articles, 10 fit the inclusion criteria and were further assessed for eligibility (Figure 2): 3 retrospective studies [12,23,24], 6 prospective studies [25,26,27,28,29,30] and 1 cross sectional study [13]. The overall characteristics of included studies are listed in Table 1. 

Overall, the risk of bias of RCTs was deemed to be low (Figure 3), whereas the risk was moderate or critical for nonrandomized studies in the measurement of outcomes and the selection of the reported results (Figure 4). 

### 3.1. Efficacy Data in Retrospective Studies

St Sauver et al. [12] conducted a retrospective, population-based cohort study of 729 men, followed for a median of 13.8 years. Statins users had a lower cumulative incidence of moderate/severe LUTS, and decreased maximum flow rate and BPH compared to nonusers. A longer duration of statins use was associated with a decreased risk of developing moderate/severe LUTS (*p* < 0.001). The strength of this study is the high number of patients enrolled and the long follow-up period (about 17 years), although they have considered only white men (the results may not be generalizable to men of other ethnicities) and patients could not use the same dose of statins.

In the study from Lee et al., the authors evaluated the effects of statins with alpha-blockers and 5-ARI, investigating changes at the time of initial treatment and after 1 year. Patients were divided into four groups according to medication by Lee et al. [23]: group A with alpha-blockers, group B alpha-blocker + statin, group C alpha-blocker + dutasteride, group D alpha-blocker + dutasteride + statin. After 1 year of follow-up, PSA (prostate specific antigen) decreased in groups B, C and D (*p* < 0.001). In addition, the PSA reduction in group B was statistically significant when compared to group A (*p* < 0.001). PSA reduction was higher in the C and D groups (dutasteride groups) than in the B group. However, the difference in PSA reduction between groups C and D was not statistically significant (*p* = 0.682). Interestingly, PV (prostate size was measured with a biplanar transrectal ultrasonography probe and volume was measured by using formula for a prostate ellipse, width × length × height × 0.52) also showed a reduction in groups B, C and D (*p* < 0.001), with a statistically significant difference between groups A and B (*p* < 0.001), but not between groups C and D (*p* = 0.762). Statin users and nonusers were also compared, excluding dutasteride users. Statin users showed a surprising higher likelihood of PSA reduction (HR 12.4, 95%CI 5.1–33.2%; *p* < 0.001) and PV reduction (HR: 14.8; 95%CI 5.8–37.6%; *p* < 0.001) after 1 year than nonusers. In conclusion, the study demonstrated that serum PSA, PV, and total cholesterol were decreased in BPH patients taking statin medication for 1 year compared with the group taking α-blocker alone. Additionally, the effect of 5ARI on the reduction of PSA and PV was not affected by adding a statin. Statin administration reduced PSA and PV in BPH patients. This infers that statin medication could improve LUTS in addition to preventing cardiovascular disease, which might play a role in the chemoprevention of prostate cancer.

This study had some limitations. First, the effect of alpha-blockers was considered to be the same, and they did not subdivide into categories for statin.

Another retrospective study was conducted by Davis et al. [24], evaluating the risk of gonadal and sexual dysfunction, including BPH (although BPH was defined as any occurrence of an ICD-9-CM code during any inpatient or outpatient healthcare encounter during the study follow-up period), among statin users and non-users. In their study, in a population of 20,731 patients, statin use was not significantly associated with an increased or decreased risk of BPH (OR 1.08; 95%CI 0.97–1.19), erectile disfunction (OR 1.01; 95%CI 0.90–1.13), infertility (OR 1.22; 95%CI 0.66–2.29), testicular dysfunction (OR 0.91; 95%CI 0.73–1.14), psychosexual dysfunction (OR 1.02; 95%CI 0.92–1.15), or all psycho-gonadism disorders (OR 1.03; 95%CI 0.94–1.14).

However, in 14,354 patients with no Charlson comorbidities, statin use was significantly associated with an increased risk for BPH (OR 1.21; 95%CI 1.08–1.35), but not for sexual disorder. However, the presence or absence of hyperlipidemia could have affected study results, as hyperlipidemia was likely more common among statin users. Moreover, other drugs that may affect prostate and sexual function, such as phosphodiesterase inhibitors and alpha blockers, were not accounted for.

### 3.2. Efficacy Data in Prospective Studies

Zhang et al. [25] evaluated the effects of simvastatin and atorvastatin in elderly males with BPH and MetS. A total of 124 patients were randomized into three groups (41 patients to a control group, 43 to the simvastatin group and 40 to the atorvastatin group) for a follow-up of 12 months. They found a significant IPSS difference between statins groups compared to the control group (*p* = 0.012), without any difference between the two statin drugs. Moreover, both atorvastatin and simvastatin treatment significantly reduced PV. However, the mean PV in the simvastatin group was smaller than in the atorvastatin group after 12 months. They also evaluated testosterone levels, without statistical difference between the three groups. The most important limitation of this study is that the patient number in each group is relatively low. 

Cakir et al. [26] compared the efficacy of statins and alpha-blockers in the treatment of LUTS/BPH. Three hundred patients were enrolled and randomized into three groups of one hundred patients: alpha-blocker only, statin only without alpha-blockers, combined therapy. Pre-treatment and post-treatment PV were not significantly different in the alpha-blocker group (HR 1.12; 95%CI −1.95, 4.21, *p* < 0.46); instead, PV was significantly lower in patients taking statin (HR 3.16; 95%CI 2.67, 3.66, *p* < 0.0001) and combined therapy (HR 3.16; 95%CI 2.59–3.73, *p* < 0.0001) at the end of the trial. Maximum urinary flow rate (Qmax), IPSS, postvoid residual urine volume (PVR) and QoL significantly changed in all three groups.

However, combined therapy was found to be superior in improving Qmax (HR −4.74; 95%CI −5.18, −4.30, *p* < 0.0001 vs. HR alpha blocker −3.16 vs. HR statin −1.55), IPSS (HR 8.38; 95%CI 7.35, 9.41, *p* < 0.0001 vs. HR alpha blocker 4.45 vs. HR statin 4.45) and QoL (HR 2.81; 95%CI 1.82, 2.10, *p* < 0.0001 vs. HR alpha blocker 1.96 vs. HR statin 1.96). The strength of the study is the randomization and the number of patients, evaluating not only statins vs. placebo, but also combined therapy with alpha-blockers. The weakness of this study is the duration of 6 months, and the absence of reporting any side effects of statins therapy.

In a 2018 study by Shih et al. [27], 7961 patients were identified with a new diagnosis of hyperlipidemia and randomized into four groups, 1604 patients with statin use >365 days, 813 patients with statin use 181–365 days, 739 patients with statin use 91–180 days, 713 patients with statin use 31–90 days and 4092 patients without using statins, as the control group. 

The risk of having BPE progression (a BPE diagnosis plus use of alpha-blockers or 5-alpha reductase inhibitors, or receiving transurethral resection of the prostate) in the cohort with statin use >365 days was significantly lower than in the control cohort (HR 0.66; 95%CI 0.54–0.8, *p* < 0.001). Instead, the risk of having BPE progression in the other statin groups did not significantly differ from the control cohort. Moreover, the trend analysis revealed that the effect of statin use on decreasing BPE progression was related to the length of statin intake (*p* < 0.001). Shih et al. evaluated statins’ effect on BPE progression in relation to the duration of therapy (31 days minimum to >365 days). Although this study shows beneficial statin effects on BPH progression, a limitation of the study is represented by the lack of a subdivision via different statins. Moreover, the authors have studied the BPH progression without reporting baseline levels of BPH.

In one of the four prospective studies, Mondul et al. [28] prospectively evaluated the impact of statin drugs use on LUTS incidence and progression, in the Health Professionals Follow-up Study from 1992 to 2008. Through questionnaires every 2 years, data about LUTS were recorded from 51,529 males (dentists, optometrists, osteopaths, podiatrists, pharmacists, and veterinarians). For LUTS incidence, after multivariable adjustment, the hazard ratio (HR) for current statin use and incidence of LUTS was greater than 1. The hazard ratio was higher for moderate or worse LUTS (IPSS ≥ 15; HR 1.11; 95%CI: 1.04–1.19) than for modest or worse LUTS (IPSS ≥ 8; HR 1.04; 95%CI 0.99–1.10). The hazard ratio did not increase with duration of use.

For LUTS progression, after multivariable adjustment, the hazard ratios for current statin use, and progression to moderate or worse LUTS or severe LUTS (IPSS ≥ 20), were slightly greater than 1. For moderate of worse LUTS, HR was 1.03, and 95%CI was 0.94–1.13; for severe LUTS, HR was 1.06 and 95%CI was 0.92–1.13. The hazard ratio did not increase with duration of use.

Moreover, in their analysis they compared men using a statin with or without hypertension medication with men using only hypertension medication. They observed that men using a statin drug and hypertension medication had essentially the same risk of LUTS as men only using hypertension medication, and men who were not using either medication had a lower risk of LUTS than men taking hypertension medications.

In this large prospective cohort of older men, statin drug use did not protect against LUTS incidence or progression. 

In a clinical study, Stamatiou et al. [29] divided 33 patients into two groups, lovastatin 80 mg daily and finasteride 5 mg daily versus only finasteride 5 mg daily. The IPSS and PV change from baseline to end point was statistically significant in both groups, while the change in mean PSA was statistically significant only in the lovastatin group (*p* = 0.00 vs. *p* = 0.02). After 4 months, there was no difference between the two groups in terms of IPSS (*p* = 0.69), PV (*p* = 0.90) and PSA (*p* = 0.16). Thus, short-term lovastatin treatment does not seem to have any additional effect on IPSS, TPV and PSA in men with BPE treated with finasteride. This 2008 study was one of the first clinical studies developed to understand the effect of statin on LUTS/BPH. Unfortunately, this trial lasted only 4 months and the maximum effect of finasteride is often not seen until after 6 months of therapy.

In a phase 2, double blind, randomized clinical study, Mills et al. [30] divided 350 patients into two groups, atorvastatin vs. placebo, with a follow-up of 26 weeks. There was no difference between atorvastatin and placebo on the primary end point, mean change in IPSS after 26 weeks, with a statistically significant difference in IPSS responder rates (≥25% reduction in IPSS vs. baseline). There was not statistical difference between the two groups on the secondary end points of PV, transition zone volume (TZV), Qmax, and PSA. The majority of the patients (91.1%) completed the study; nine patients (2.6%) discontinued because of adverse events. The majority of adverse events were mild or moderate. Only one patient reported a serious adverse event (chest pain and rhabdomyolysis), attributed to atorvastatin.

At the end, erectile function, with the International Index of Erectile Function Questionnaire (IIEF) questions 3 and 4, which evaluated the strength and durability of the erection, was also assessed at baseline and at the end of the study. This study showed no statistical difference between the atorvastatin and placebo groups. The design of this study was a double-blind, randomized, placebo-controlled clinical study to evaluate the effect of atorvastatin on BPH. They also reported side effects of atorvastatin, and a consideration about statins and erectile function. However, the authors considered only a period of 6 months.

### 3.3. Efficacy Data in Cross Sectional Study

Hall et al. [13] evaluated LUTS, voiding and storage symptoms through an interviewer-administered questionnaire with the AUA symptoms index among men and women. In this paper, statin use had no association with LUTS among women and younger men (<60 years); instead, there was an association between statins and older men (>60 years). They observed a significant inverse association (suggestive of a protective effect) for voiding (OR 0.23; 95%CI 0.08–0.66), storage (OR 0.24; 95%CI 0.11–0.56) and overall LUTS (OR 0.15; 95%CI: 0.05–0.44). Using a questionnaire is surely a limitation of this study for the subjectivity of answers. Moreover, cross-sectional study evaluates statin use and LUTS at one point in time, and cannot determine cause and effects. Regardless, the authors evaluated statin use on males and females to identify any confounding factors of statin use between men and women. At last, statin use had an association among old men, but not among women and younger men.

## 4. Discussion

Beside reducing cholesterol levels, statins also have anti-inflammatory and anti-apoptotic roles [16], involving also prostate tissue. In fact, statins increase the apoptosis of the prostatic epithelium and stroma [31], and these proprieties could be a protective side effect related to BPE and LUTS. 

In the present study, we systematically reviewed the literature investigating the association between BPE and the use of statin drugs. 

A meta-analysis by Yang et al. [32] was published on April 2020 about the effect of statins on BPH and LUTS. Unlike our systematic review, Yang et al. [32] included one abstract and two Chinese papers, studies that did not meet the inclusion criteria. Moreover, we have added two more studies, Lee et al. [23] and Cakir et al. [26]. 

All the studies, except those of Mills et al. [30], Stamatiou et al. [29] and Mondul et al. [28], suggest the protective role of statins toward BPE, reducing PV or PSA, or improving Qmax or IPSS/AUA score. 

Mills et al. [30] determined that the effect of atorvastatin on Rho/Ras isoprenylation is insufficient to mediate a clinically relevant impact on abnormal benign prostate and bladder morphology over a period of 6 months. 

In Stamatiou et al.’s study [29], there was a difference between the two groups only in the change in mean PSA from baseline to end point, although there was no difference between the two group in IPSS (*p* = 0.69), PV (*p* = 0.90) and PSA (*p* = 0.16) after 4 months of treatment. However, it could be supposed that statins would have an effect only after the maximum finasteride effect had occurred (minimum of 6 months). 

Moreover, it is possible that statins, acting on prostate growth and apoptosis, similarly to 5-alpha reductase inhibitor (5-ARI) drugs, need a longer period time to manifest results in terms of BPE.

In fact, Shih et al. [27] considered the duration of statins therapy and showed the potential of long-term statin use (more than 365 days) to reduce risks of BPE progression, and how the protective effect of statins is related to the treatment duration.

In this regard, only this study assessed this important aspect differently, and although specific recommendations cannot be given with certainty, the data suggest that longer statin prescription (1 year) is more beneficial than shorter duration. 

It is also important to discuss the potential role of confounding factors, such as diabetes, which may affect or even influence the efficacy of statin. 

In this context, for example, Shih et al. [27] evaluated the impact of DM and other confounding factors without demonstrating any influence. Unfortunately, other included studies did not report such data. 

Mondul et al.’s study [28] was a large prospective cohort study of 51,529 patients. They observed that statins did not protect against LUTS incidence or progression. A possible explanation is that, unlike other trials, this study included health professionals, who are usually younger and more prone to periodic check-ups and prevention. 

The same considerations could be made for Davis et al.’s study [24]. In fact, in a population of patients with no Carlson comorbidity, statin use may be a more important risk factor for BPH among healthier men. Statin users suffer from hypercholesterolemia, and usually have MetS, a detrimental factor for LUTS. 

Russo et al. [10] showed that the presence of MetS worsened the symptoms of BPH, including PV. Additionally, De Nunzio et al. [11] and Sebastianelli et al. [33] underlined how MetS, with systemic inflammation, influence the response to the medical treatment of BPE. 

Thus, in addition to their direct role in prostate growing, statins, improving MetS parameters and reducing systemic inflammation, might offer a better answer to LUTS therapy. 

LUTS due to prostate enlargement, and improving with statins, was demonstrated by Hall et al.’s study [13]. In fact, statin drugs use had no association with LUTS among women and younger men (<60 years); instead there was an association between statins and older men (>60 years). Beside these considerations, we might raise some questions regarding the impact of greater prostate volume on potential statin efficacy. Previously, it was demonstrated that some MetS features (hyperinsuline- mia, dyslipidemia) contribute to inflammation-driven prostatic overgrowth [34,35,36]. For this reason, although we do not have clinical data from the literature, we may suppose that the efficacy of statins can be consistently influenced by prostate volume. 

Regarding statins safety, the American Heart Association have stated that, despite statins’ serious adverse effects, such as myopathy, new diagnoses of diabetes mellitus and hemorrhagic strokes, in the patient population to whom statins are recommended by current guidelines, the cardiovascular benefits far outweigh any safety concerns [37]. Mills et al. [30] evaluated statin’s side effects, reporting only one serious adverse effect in a populations of 176 enrolled patients. 

Additionally, lower levels of testosterone were reported as an adverse effect of statin therapy [19,38]. In this systematic review, Mills et al. [30] and Zhang et al. [25] evaluated erectile function and testosterone levels. 

Mills et al.’s prospective study [30] considered sexual function in patients with BPH, studying mean change in scores for IIEF questions 3 and 4. They showed no difference between the atorvastatin and placebo groups. 

Davis et al. [24] performed a retrospective study evaluating the risk of gonadal and sexual dysfunction. The results of our study suggest that statins may not be beneficial for the prevention of sexual dysfunction, nor do they increase the risk of such events. 

Moreover, there were no statistical differences in testosterone levels between the simvastatin group, the atorvastatin group and the control group in Zhang et al.’s study [25].

There are still many limitations in this systematic review. First, there are few studies on the correlation between BPH and statins, with a need for further evidence. Second, we have considered only English-language studies, so there is a possible language bias. Third, we gained heterogeneous outcomes due to the different kinds of measurements of BPH and LUTS. Fourth, the available studies of this systematic review have used different statins and different doses of statins, so it is difficult to evaluate if statins’ effects on BPH are related to a particular kind of statin, or if they are dose-dependent.

However, taking together all these findings and the potential role of statins in BPH/LUTS, we would encourage the performing of an RCT assessing its efficacy when added to low dosage PDE5i or alpha-blockers. Recent evidence, in fact, has demonstrated that statins inhibit inflammation, angiogenesis, cell proliferation, migration/adhesion, and invasion, and promote apoptosis [39]. Harshman et al. showed that statins inhibit cell androgen uptake by competing for intracellular transport sites in the solute carrier organic anion transporter family, member 2B1 (SLCO2B1). Specifically, the uptake of dehydroepiandrosterone sulphate (DHEAS), a precursor of potent androgens such as dihydroxytestosterone (DHT), is dramatically reduced by statin exposure. Thus, statins may reduce intracellular androgen supply. This mechanism may provide the basis for synergy between statins and prostate disease [40].

## 5. Conclusions

Statins, with their pro-apoptotic and anti-inflammatory proprieties, showed beneficial effects on LUTS in patients with BPE/BPH in most of the trials evaluated. Data from our systematic review suggest that a long-term therapy with statins, for at least 6 months, is required to achieve significant impacts on prostate tissue and LUTS. Moreover, besides statins’ direct activity, the risk reduction of LUTS might be connected to the improvement of hypercholesterolemia and MetS.

Despite most of the evidence seeming to support a protective effect of statins on the prostate, improving BPE and LUTS, the results are still controversial, not allowing a definitive indication of the actual effectiveness and clinical meaningfulness. More prospective studies, with multivariate analysis and evaluating wide populations, are needed for a better comprehension of this relation.

## Figures and Tables

**Figure 1 jcm-10-00416-f001:**
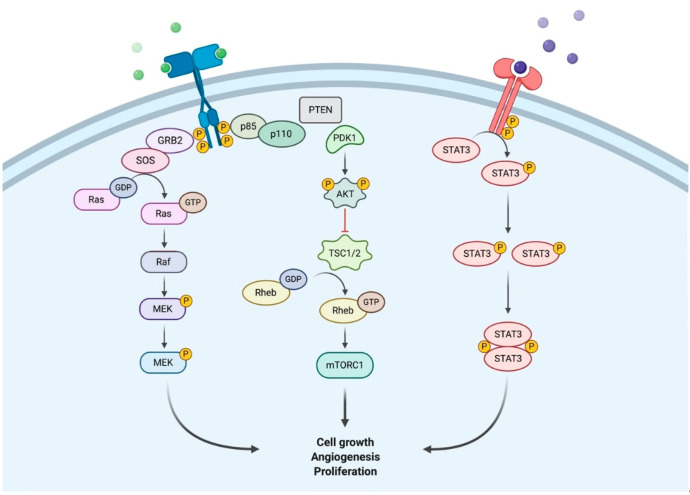
Insights representation of the activity of Ras pathways. Green and purple circles represent the mediator.

**Figure 2 jcm-10-00416-f002:**
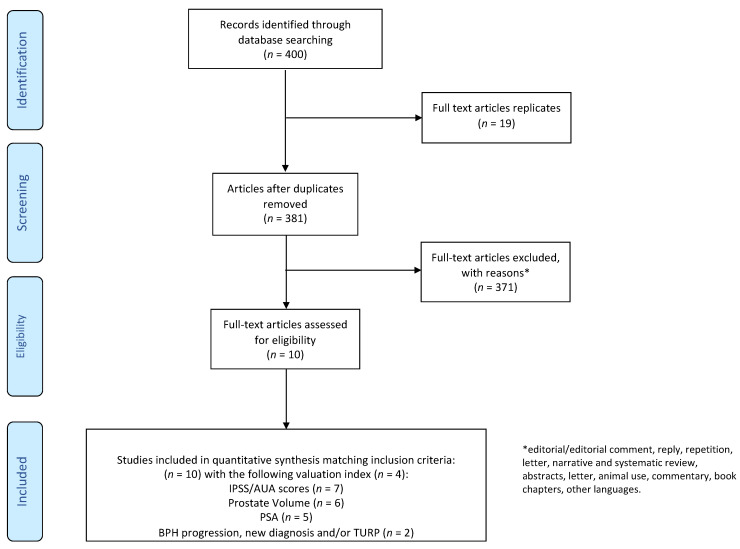
PRISMA (Preferred Reporting Items for Systematic Reviews and Meta-Analyses) 2009 flow diagram.

**Figure 3 jcm-10-00416-f003:**
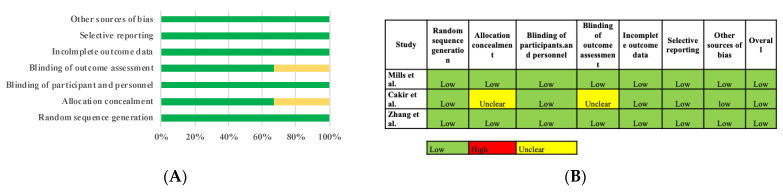
(**A**) Risk of bias assessment in randomized studies and (**B**) summary of randomized studies: review authors’ judgements about each risk of bias item for each included study. Green = low risk of bias; yellow = unclear risk of bias; red = high risk of bias.

**Figure 4 jcm-10-00416-f004:**
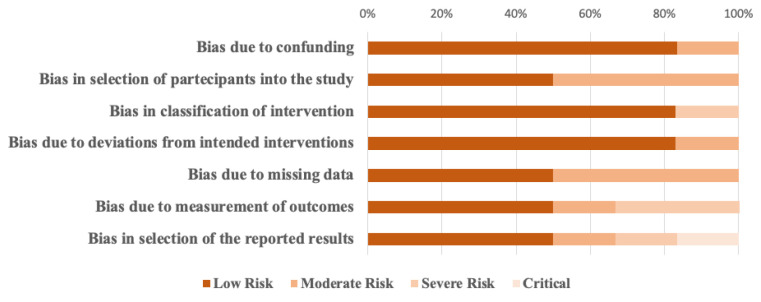
Risk of bias assessment in nonrandomized studies according to ROBINS-I tool. ROBINS-I = risk of bias in nonrandomized studies of interventions.

**Table 1 jcm-10-00416-t001:** List of included studies with main characteristics.

Study	Year	Design	Valuation Index	Groups Division	N. Patients	Follow-up	IPSS/AUA Scores	*p*-Value	PSA	*p*-Value	PV	*p*-Value
							From baseline		From baseline		From baseline	
Mills et al. [30]	2007	RCT	IPSS, PV, Qmax and PSA	80 mg Atorvastatin daily	176	26 weeks	−4.2	0.263	−0.1	0.235	−2	0.654
				Control	174		−3.5		0		−2.4	
							Pre-treatment vs post-treatment			Pre-treatment vs post-treatment		
Zhang et al. [25]	2015	Prospective cohort study	PV, PSA and IPSS	40 mg Simvastatin daily	43	1 year	8.49 ± 4.59	<0.05	1.97 ± 1.92	0.027	37.39 ± 17.47	0.01
				20 mg Atorvastatin daily	40		8.15 ± 5.49		1.87 ± 1.74		44.78 ± 20.97	0.05
				Control	41		11.02 ± 7.31		2.02 ± 1.91		48.31 ± 18.29	
							From baseline		From baseline		From baseline	
Cakir et al. [26]	2018	Prospective cohort study	Qmax, PV, IPSS, QoL, PVR	Tamsulosin	100		19.2/14.6	0.001		HR = 1.12 95%CI–1.95–4.21	0.46	
				Atorvastatin	100		19.0/14.8	0.001		HR = 3.16 95%CI 2.67–3.66	0.001	
				Tamsulosin + Atorvastatin	100		17.1/8.7	0.001		HR = 3.16 95%CI 2.59–3.73	0.001	
							BPE progression					
Shih et al. [27]	2018	Prospective cohort study	BPE progression (new diagnosis of BPE plus drugs use or TURP)	Statin user >1 year	1604		HR = 0.7 95%CI 0.58–0.85	<0.001				
				Statin user > 181–365 days	813		HR = 0.98 95%CI 0.77–1.27	0.899				
				Statin user 91–180	739		HR = 0.99 95%CI 75–1.31	0.944				
				Statin user 31–90	713		HR = 1.07 95%CI 0.8–1.42	0.672				
				Control group	4092		HR = 1					
Stamatiou et al. [29]	2008	Prospective cohort study	IPSS, PV, PSA	Finasteride 5 mg + Lovastatin 80 mg daily	18	4 months	from 14 to 7.5	*p* = 0.00	from 2.87 to 1.89	*p* = 0.00	from 58.7 to 46.8	*p* = 0.00
							Incident LUTS		LUTS progression			
Mondul et al. [28]	2013	Prospective cohort study	Association between severity of LUTS and statin	Statin user	4238	16 years	HR = 1.04 95%CI 0.99–1.10 IPSS ≥ 8 HR = 1.11 95%CI 1.04–1.19 IPSS ≥ 15		HR = 1.03 95%CI 0.94–1.13 IPSS ≥ 15 HR = 1.02 95%CI 0.92–1.13 IPSS ≥ 20			
				Control	13,644		HR = 1		HR = 1			
Hall et al. [13]	2011	Cross sectional	Association between severity of LUTS and statin	Statin user	231		OR = 0.23 (0.08–0.66) (Voiding symptoms) OR = 0.24 (0.11–0.56) (Storage voiding) OR = 0.15 (0.05–0.44) (LUTS)					
				Control	319							
							AUA symptoms score > 7				N of patients PV > 30 mL	
St Sauver et al. [12]	2010	Retrospective cohort study	AUA symptoms score	Statin user	729	15.5 years	103 (40.08%)	0.004			21 (31.82%)	0.34
				Non statin user	1718	11.9 years	701 (49.93%)				133 (38%)	
									Increasing PSA		Increasing PV	
Lee et al. [23]	2013	Retrospective cohort study	PV and PSA	Statin user	142	1 year			29 (20.4%)	<0.001	36 (25.4%)	<0.001
				Non statin user	281				238 (84.7%)		261 (92.9%)	
Davis et al. [24]	2015	Retrospective cohort study	Occurrence of benign prostatic hypertrophy (BPH)	Statin user	3542	7 years					1.224 (34.6%)	OR 1.08; 95%CI 0.97–1.19
				Control	10,812						1.527 (14.1%)	

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
