# Peer review of "The Investigative Role of Statins in Ameliorating Lower Urinary Tract Symptoms (LUTS): A Systematic Review"

_jcm, 2021, doi:10.3390/jcm10030416_

Round 1

Reviewer 1 Report

The authors report on a meta-analysis of reports analyzing the effects of statins on benign prostatic enlargement. This paper is very well written and very comprehensive. As one addition, the authors could suggest a trial design to compare statins to established BPH treatments such as low-does PDE5I or alpha-blockers. They could also further mention how statins may be additive to these drugs.

Author Response

Response to Reviewer 1 Comments

Point 1: The authors report on a meta-analysis of reports analysing the effects of statins on benign prostatic enlargement. This paper is very well written and very comprehensive. As one addition, the authors could suggest a trial design to compare statins to established BPH treatments such as low-does PDE5I or alpha-blockers. They could also further mention how statins may be additive to these drugs. 

Response 1:

We would like to thank you for your precious support and suggestions. We have modified the discussion by adding the following sentences:

However, taken together all these findings and potential role of statin on BPH/LUTS, we would encourage to perform a RCT assessing its efficacy when added to low dosage PDE5i or alpha-blockers. Recent evidences, in fact, have demonstrated that statins inhibit inflammation, angiogenesis, cell proliferation, migration/adhesion, and invasion, and promote apoptosis. Harshman et al showed that statins inhibit cell androgen uptake by competing for intracellular transport sites at solute carrier organic anion transporter family, member 2B1 (SLCO2B1). Specifically, uptake of dehydroepiandrosterone sulphate (DHEAS), a precursor of potent androgens such as dihydroxytestosterone (DHT), is dramatically reduced by statin exposure. Thus, statins may reduce intracellular androgen supply. This mechanism may provide the basis for synergy between statins and prostate disease.  

Reviewer 2 Report

This is a systematic review summarizing 10 studies that have explored how statin use may associate with risk of benign prostatic hyperplasia or lower urinary tract symptoms. Generally I find the manuscript an informative introduction to current state-of-the-art regarding the topic. Nevertheless, following issues need to be addressed:

  • The included studies comprise two clinical trials (one of which is randomized), seven cohort studies and one cross-sectional study. The studies are now grouped to retrospective and prospective studies, where prospective cohort studies are handled together with clinical trials in the latter group. It would be more informative to group the clinical interventional studies (RCTs) and observational studies (cohort studies and cross-sectional) together. This is because in the interventional studies timing of statin exposure is well-defined, whereas in observational studies duration of statin use is usually unknown. Additionally, the study by Mills is a randomized clinical trial, where randomization mitigates confounding by background factors, which is not the case for any other included study.
  • Table 1: The study by Stamatiou etl al 2008 is not a randomized clinical trial, as the intervention was not randomized but instead chosen by the participants' lipid level. Therefore the study design is not RCT
  • 3.1. Efficacy data in retrospective studies, starting on line 151: The study by Lee et al is introduced and discussed in much less detail than other included studies. This should be corrected.
  •   Although this is not a meta-analysis, a graphical representation comparing results between different studies would help the reader. Forest plot of the risk estimates would work well.
  • You should discuss which studies were able to accurately determine length of statin use, as this likely varies a lot between the studies. Also discuss which studies were able to take into account cholesterol level and underlying diabetes, as these may be confounders.
  • Also discuss whether prostate volume modifies the risk association between statin use and BPH outcomes.

Author Response

Response to Reviewer 2 Comments

Point 1: This is a systematic review summarizing 10 studies that have explored how statin use may associate with risk of benign prostatic hyperplasia or lower urinary tract symptoms. Generally, I find the manuscript an informative introduction to current state-of-the-art regarding the topic. Nevertheless, following issues need to be addressed:

  • The included studies comprise two clinical trials (one of which is randomized), seven cohort studies and one cross-sectional study. The studies are now grouped to retrospective and prospective studies, where prospective cohort studies are handled together with clinical trials in the latter group. It would be more informative to group the clinical interventional studies (RCTs) and observational studies (cohort studies and cross-sectional) together. This is because in the interventional studies timing of statin exposure is well-defined, whereas in observational studies duration of statin use is usually unknown. Additionally, the study by Mills is a randomized clinical trial, where randomization mitigates confounding by background factors, which is not the case for any other included study.

Response 1:

We would like to thank you for your precious support and suggestions.

We have modified the table according to your suggestions.

Point 2: Table 1: The study by Stamatiou et al 2008 is not a randomized clinical trial, as the intervention was not randomized but instead chosen by the participants' lipid level. Therefore, the study design is not RCT

Response 2:

Yes, we apologize for the misunderstanding. We have modified the table accordingly.

Point 3: 3.1. Efficacy data in retrospective studies, starting on line 151: The study by Lee et al is introduced and discussed in much less detail than other included studies. This should be corrected.

Response 3:

We have updated the discussion according to the part of Lee et al. 

Point 4: Although this is not a meta-analysis, a graphical representation comparing results between different studies would help the reader. Forest plot of the risk estimates would work well.

Response 4:

We agree with your observation and also this point was internally assessed in our research group. Unfortunately, data, as they are reported in the studies, are not possible to be merged since the report different outcomes. Furthermore, if we consider that retrospective studies cannot be included, the number of studies is few to perform a meta-analysis (at least 10 studies should be included).

We hope that our paper gives more insights in this field in order to perform well conducted observational studies.

Point 5: You should discuss which studies were able to accurately determine length of statin use, as this likely varies a lot between the studies. Also discuss which studies were able to take into account cholesterol level and underlying diabetes, as these may be confounders.

Response 5:

We have highlighted changes in the discussion regarding these questions.

Point 6: Also discuss whether prostate volume modifies the risk association between statin use and BPH outcomes.

Response 6:

We have changed the discussion accordingly.

LUTS due to prostate enlargement and improving with statins is showed by Hall et al. study 13. In fact, statin drugs use had no association with LUTS among women and younger men (<60 years), instead there was an association between statins and older men (>60 years). Beside these considerations, we might arise some questions regarding the impact of greater prostate volume on potential statin efficacy. Previously, it has been demonstrated that ome MetS features (hyperinsuline- mia, dyslipidemia) contribute to the inflammation- driven prostatic overgrowth 34–36. For this reason, although we do not have clinical data from literature, we may suppose that the efficacy of statin can be consistently influenced by prostate volume

This manuscript is a resubmission of an earlier submission. The following is a list of the peer review reports and author responses from that submission.

Round 1

Reviewer 1 Report

The authors report on a systematic review of papers addressing the role of statins for the improvement of lower urinary tract symptoms. They found that in the majority of studies, statins have shown to improve these symptoms.

The authors provide a clear and concise review with stringent inclusion criteria. The Discussion is succinct and the results are intriguing. The authors address the discrepancy among study results very well. This paper would be an addition to the existing literature.

Reviewer 2 Report

Authors have made substantial efforts to review the current literature on the possible association between statin use and change of LUTS. Although mentioned as a systematic review, some aspects of systematic reviews are not reported.

I have a number of concerns with their work, that are also applicable to a previously published systematic reviews from this study group. I would strongly advice the authors to invite an epidemiologist that could help interpreting the included studies. When studying a causal association, basic knowledge about appropriate study designs is needed. For each study type included, the major flaws need to be recognized. I feel that this knowledge is lacking and interpretation of studies is flawed, or that flawed studies are misinterpreted. This is particularly illustrated by including a cross sectional study in this review. Importantly, authors should also be critical concerning referring to other cross sectional studies (in the introduction and discussion). Such studies could show associations, but by no means could provide information on causality. In many sentences, authors suggest a causal relationship despite referring to cross sectional studies.

For other study types, the accuracy of the data-collection, including identification of patients and exposures (in this case statin use) need to be explained.

Next, an epidemiologist/methodologist could help apply the correct terms, e.g. on efficacy (impossible to study in retrospective cohorts), and study designs of included studies (see further).

The significance of the outcomes remains unclear. Authors do not add any information on the clinical relevance. Please take the minimal clinically important difference (MCID) of outcomes into account.

Additional comments

Introduction

Please add that the association described by Pashootan was based on a cross-sectional study only. Information about the adjustment for age (amongst others) is also relevant.

The systematic review by De Nunzio et al also included cross sectional studies.

Methods

Please explain how you decided upon pooling the outcomes, or explain why pooling of the data was not suitable. Please apply appropriate ways for data synthesis in case of not pooling the data.

Results

Risk of bias of observational studies should be presented differently. Please show for each included study the separate outcomes, as only this will provide insight in the quality of different studies.

Retrospective studies also by definition have enormous problems in the selection of patients, completeness of data and large confounding. The current review suggests that bias due to confounding is not relevant. I cannot believe this (see my general comment above).

Please provide more details to the definition of study designs (Table 1). RCTs are also prospective studies, and clinical studies may also include observational studies. Please use:

retrospective cohort study

prospective cohort study

RCT

Why do authors add p-values for baseline comparisons for the RCTs? This makes no sense, because by definition, differences are due to coincidence.

Why is only a p-value mentioned for the St Sauver study PSA values, but not the outcomes?

Table 1 also contains hazard ratios. Why? This is not part of the description of study characteristics.

Please restructure Table 1 and provide a decent footnote explaining the abbreviations. It is helpful for the readers when type of studies are grouped. It is also helpful to explain how participants were identified and data were collected, in more detail. Don't include the outcomes of interest in this table.

Please do not report p-values in the text without additional information on the outcomes that were compared. P-values itself don’t give any information on the clinical relevance of the differences.

Please add information on PV measurement in the study performed by Lee et al.

Please add to the text that the study of Zangh et al was an RCT. So patients were "divided" means patients were randomized. Same for the other RCTs.

Discussion

As I feel that the current way this work is presented, I am unable to judge the consistency of the results, which makes it impossible to properly judge the discussion section. It seems that authors themselves have some confirmation bias, as - despite a number of negative findings - the positive findings get much more attention.